# Agroecological Management and Increased Grain Legume Area Needed to Meet Nitrogen Reduction Targets for Greenhouse Gas Emissions

**Geoffrey R. Squire \*** , **Mark W. Young and Cathy Hawes**

James Hutton Institute, Dundee DD2 5DA, UK
\* Correspondence: geoff.squire@hutton.ac.uk

**Abstract:** The nitrogen applied (N-input) to cropping systems supports a high yield but generates major environmental pollution in the form of greenhouse gas (GHG) emissions and losses to land and water (N-surplus). This paper examines the scope to meet both GHG emission targets and zero N-surplus in high-intensity, mainly cereal, cropping in a region of the Atlantic zone in Europe. A regional survey provides background to crops grown at an experimental farm platform over a run of 5 years. For three main cereal crops under standard management (mean N-input 154 kg ha$^{-1}$), N-surplus remained well above zero (single year maximum 55% of N-input, five-year mean 27%), but was reduced to near zero by crop diversification (three cereals, one oilseed and one grain legume) and converted to a net nitrogen gain (+39 kg ha$^{-1}$, 25 crop-years) by implementing low nitrification management in all fields. Up-scaling N-input to the agricultural region indicated the government GHG emissions target of 70% of the 1990 mean could only be met with a combination of low nitrification management and raising the proportion of grain legumes from the current 1–2% to at least 10% at the expense of high-input cereals. Major strategic change in the agri-food system of the region is therefore needed to meet GHG emissions targets.

**Keywords:** nitrogen fertilizer; nitrogen surplus; greenhouse gas emissions; emissions reduction; agro-ecology; cereals; oilseed; pulse; legume; low nitrification





## 1. Introduction

Nitrogen has the potential to increase crop and pasture production, alleviate hunger and improve human nutrition, but also to pollute the environment if used inefficiently, especially when supporting livestock in preference to food crops. The positive and negative effects of nitrogen were known at the start of 20th century intensification [1]. Over the three decades from 1960 to 1990, a five-fold rise in nitrogen fertiliser in global agriculture supported a doubling of the human population on a stable area of arable land [2]. However, despite all prior warnings, more nitrogen was routinely applied to crops than was needed, much more nitrogen was used to support livestock than food crops, especially in developed countries, and the 'leakage' of nitrogen from agricultural land brought major harm to ecosystems and human health [3–5].

After 1990, the rise in yield of the main crops stalled in many parts of the world, followed by more than a quarter of a century of near-level, or at best, slightly rising, output [6–8]. Some inputs were regulated—for example, less phosphate fertiliser was applied in many regions when it was realised that the excess given in the early 20th century had been stored in soil and could be gradually released [9,10]. In contrast, high usage of nitrogen fertiliser continued, causing further pollution to water [11] and greenhouse gas (GHG) emissions to the atmosphere [12,13]. The full extent of environmental harm is yet to be realised, since agrochemicals have delayed effects through accumulation in the soil and subsequent release [14–16]. The imperative in developed agriculture is to minimise nitrogen usage and losses, but progress has been variable. In Europe, for example,

policies to reduce nitrogen and other agricultural inputs [17,18] were effective in many target areas across north-west Europe [19,20], but inputs have since levelled or increased in many countries [21]. The costs to society of the damage due to fertiliser nitrogen may now be greater than the benefits brought by raised agricultural production [5].

Current scientific understanding has defined the problem and its potential solutions. The biophysical pathways of nitrogen dynamics and loss are well researched. Applied nitrogen fertiliser is rapidly converted by nitrification in soil to forms that are highly mobile and hence prone to movement out of the rooting zone before they are taken up by plants [4,22]. Practices to reduce losses—sometimes termed 'low-nitrification management'—include crop diversification, minimum soil tillage, adding nitrification inhibitors to soil and matching N application with demand [23–25]. Recommended practice comprises a 'portfolio' of approaches [23], rather than a single change in management, to cover the variety of conditions across soils, crops and climates [25]. The single most effective means to reduce mineral nitrogen inputs to arable systems is to include legumes as grains and forages that acquire their nitrogen through biological fixation [26,27]. However, low nitrification management is not widely applied, with the consequence that legumes remain minor crops in most of Europe and nitrogen losses have continued on a large scale [28–30]. The inertia in agri-food systems and lock-in to existing supply chains results in continued pollution. In the UK, for example, where the present study is based, the latest government survey estimated nitrogen surplus or loss from agriculture as a whole was 46% of nitrogen inputs [31].

Though the problems with nitrogen fertiliser are global, solutions need to be tailored to local crops and conditions [25]. A major case study was therefore initiated in a region of lowland Scotland that supports mixed farming of high productivity, comprising both grassland and arable land, typical of the Atlantic climatic zone [32]. After 1990, both nitrogen and phosphorus inputs were reduced proportionately in grassland, but only phosphorus was reduced in arable (Figure 1). The resistance to reducing nitrogen in arable lies in the need for repeated application to satisfy a high nitrogen requirement in cereal grain. Pressure has intensified in recent years from conservation bodies and government to counter the contribution of nitrogen fertiliser to habitat degradation and greenhouse gas emissions [33–35]. Targets for emissions from agriculture as a whole have been set by the government to reach at least 70% of the 1990 base value by 2032 [34]. A reduction in this magnitude is indicated on Figure 1 as a nominal target for nitrogen input.

The study had three phases. First, to provide a context for the main experiment, trends of nitrogen inputs and losses in the main grain crops were defined at a regional scale from government census. Second, the scope for reducing input and surplus was assessed over five years at the Centre for Sustainable Cropping (CSC), a major 42 ha, field-scale, experimental platform [36]. The experiment measured the degree to which nitrogen inputs could be reduced by (1) agronomic improvements to soil management, and (2) crop diversification including the introduction of a grain legume, spring bean (*Vicia faba* L.) which is capable of high fixation rates [36,37]. Third, findings at the platform were up-scaled to assess whether targets could be met at regional level in the main grain-based cropping systems [38]. The paper concludes as to whether reduction targets can be met simply by modification of current practice or whether more fundamental change is needed in the agri-food system. While the example is regional in extent, the study should show the wider relevance of combining long term survey and field-scale experimental platforms.

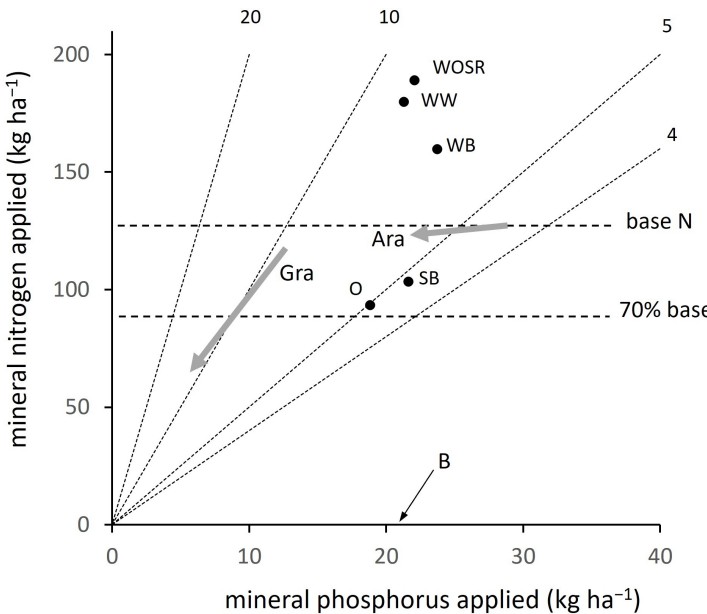

**Figure 1.** Trends, shown by grey arrows, in mineral nitrogen and phosphorus inputs for grassland (Gra) and arable (Ara) for the fertiliser census area of Scotland [39]. In each of the two trends, the beginning of the arrow is located at the mean for the five years around 1990 and the point of the arrow at the mean for the five years around 2014 (the period of the first phase of the CSC platform presented here). Numbers show the N:P ratio for guidance. Symbols show the positions for the main grain crops: winter oilseed rape (WOSR), winter wheat (WW), winter barley (WB), spring barley (SB), oats (O), and beans (B, to which no nitrogen is usually given). Horizontal lines show the mean nitrogen input to arable in 1990 (the base for emissions estimates) and 70% of the base (see text for justification).

## 2. Materials and Methods

### 2.1. Regional Context for Crops, Yield and Cropping Systems

Results from the Centre for Sustainable Cropping (CSC) are considered in relation to regional states and trends derived from government survey. Land in managed agriculture is classed as grass or arable (the latter also termed 'tillage' in some surveys). This study considers only arable data. Areas of land grown with different crops are recorded for the whole region in an annual census, whereas yield per unit area is derived separately from several sources [40,41]. The main arable crops can be classed as short-season or long-season. Short-season crops take 5–6 months from sowing to harvest, are sown in spring and harvested autumn the same year, and hence are termed *spring crops*; the main cereal types are spring barley and oats. Long-season crops take 10–11 months, are sown in autumn, remain over winter and are harvested late summer or early autumn the next year; these are termed *winter crops*, the main types of which are winter wheat, winter barley and winter oilseed rape, and all receive more fertiliser and pesticide inputs and yield more than short-season types. During the five-year period of the CSC reported here (2012–2016), arable crops in the region were grown on 588,000 ha, of which about 80% were grains: spring barley 41%, winter wheat 19%, winter barley 8%, oats 4%, winter oilseed rape 6%, and some minor cereals. Fallow, forage crops and 'root' crops made up the rest. Mean grain yields (at 14% water content) over the decade up to 2016 were 6.51 t ha$^{-1}$ for all cereals, 5.68 t ha$^{-1}$ for spring barley, 7.15 ha$^{-1}$ winter barley and 8.22 ha$^{-1}$ for wheat. In relation to indicators such as nitrogen surplus and waste (considered later), winter crops, especially wheat, are prone to damage, yield loss and incomplete harvesting (and hence reduced nitrogen uptake and offtake) by soil waterlogging due to heavy and persistent rain. The period of 5 years of the study included a year (2012) in which most crops suffered due to prolonged wetness. In the period from the start of rapid crop bulking to the latest harvest

(April to November), the climatic region (East Scotland) experienced in that year 4 of the 10 months of highest rainfall in the five years of the study and had only 12 days classed as without rain [42]. Other years were more typical of the region, whereas the weather in 2015 and 2016 supported the highest cereal yields of recent decades.

The area and yield census, and the fertiliser survey (see later), provide data averaged by crop. In practice, crops are grown in different combinations to form several distinct systems. Using census data [39], arable and arable-grass fields were previously assigned to intensity systems according to several criteria. Arable fields that exclude grass and are dominated by cereals fall into three systems: mainly spring cereals (no winter cereal); mixed winter and spring cereals; and winter cereal (no spring cereal). These three cereal-based systems are used for nominal up-scaling of the CSC platform's results to the region, based on percentage areas of 29% for spring, 55% for mixed, 16% for winter.

### 2.2. Centre for Sustainable Cropping Experimental Platform

The Centre for Sustainable Cropping (CSC), based at Balruddery Farm near Dundee, North-East Scotland (56.48 latitude, −3.13 longitude), is a long-term experimental platform comprising a 42 ha block, established in 2009 to design and test a systems approach to optimising crop yield, biodiversity and ecosystem services, while reducing environmental footprint by minimising inputs and the loss of non-renewable resources [36,43]. The CSC compares, in a split-field experiment, the *standard* agronomic practice typical of cropping systems in the region with an *integrated* cropping system. The same grain crops are grown in standard and integrated systems: three cereals (spring barley, winter barley and winter wheat) and two broadleaf crops (winter oilseed rape and spring beans). The integrated system aims for increased soil organic matter (green waste compost, cover crops, crop residue incorporation), reduced soil disturbance and surface wash (non-inversion tillage), enhanced arable biodiversity (targeted weed management, diverse field margins, IPM strategies) and reduced nitrogen (lower input, enhanced N-fixation). The practices in the integrated system with nitrogen-fixing beans are termed *agroecological management*.

The CSC incorporates two broad categories of what has been termed a 'portfolio approach' [23] to low nitrification management [22,44]. The first is crop diversification in the form of 60% cereals and 40% broadleaf grain crops in what is otherwise a cereal-dominated region. Both the broadleaf crops—oilseed rape and the grain legume spring beans—are grown in the region along with high input cereals, though the area that includes beans in the rotation is now very small. The second category includes the range of practices in the integrated treatment (listed above) that in combination (1) allow lower N fertiliser inputs, (2) reduce fertiliser loss from the soil surface, and (3) facilitate the incorporation and retention of nitrogen into soil by raising soil organic matter. The experiment can therefore assess whether nitrogen reduction targets can be met under standard management by crop diversification alone, or by a combination of diversification and integrated management. Winter oilseed rape was sown each year following a spring or winter cereal, whereas bean was sown after one or other of the winter cereals.

Data under the two management systems are presented here for five consecutive years during the platform's first phase of detailed record up to 2016, therefore constituting 25 crop-years of data for each of the standard and integrated treatments. Sampling took place in late summer when plant uptake of N was near maximal and before the crop began to senesce and potentially lose material. All of the above-ground vegetation was sampled from 1 × 0.5 m quadrats at 3 locations, approximately 40 m apart along 5 or 6 transects spaced 18 m apart in each treatment, providing 174 samples in total each year for 5 years. The material was divided into vegetative (stem and leaves) and reproductive (grain) parts, dried at 70 °C for 48 h to give dry weight per unit area. Sub-samples were analysed for % nitrogen following standard protocols using an Element Analyser, as described previously [45]. The total N in the plant was derived from dry matter and % nitrogen and presented per unit field area.

For each crop, differences between integrated and conventional treatments (declared as a factor with two levels) were tested using a general Analysis of Variance in GenStat for Windows 17th edition (VSN International Ltd., Hemel Hempstead, UK): the main variables examined were N content in the whole plant (kg ha$^{-1}$), and N-surplus derived from N-input minus N content (kg ha$^{-1}$). Residual plots were checked for conformity to a normal distribution and log-transformed where necessary.

### 2.3. Emissions Targets, Nitrogen Fertilizer, and Nitrogen Surplus

The paper addresses the question of whether fertiliser nitrogen input to crops can be reduced in line with emissions targets set by government. Greenhouse gas emissions from agriculture are calculated in relation to the base year of 1990 by the National Atmospheric Emissions Inventory [46] and summarised for the UK and Scotland by the Committee on Climate Change [34,47]. Emissions from agriculture of 9 MtCO$_2$e in 1990 had declined by 14% to 7.7 MtCO$_2$e at the time of the CSC experiment. However, early progress has not been maintained in recent years and emissions are well above the government target for agriculture overall in Scotland of 70% [34] or 60% [48] of the 1990 baseline. Nitrogen use in arable cropping is a major contributor that has changed little since 1990, but no specific reduction targets have been set for nitrogen fertiliser input. For the analysis in this paper, therefore, the baseline is assumed to be the mean 'overall application rate' of 127 kg ha$^{-1}$ averaged over the years 1988–1992 as in Figure 1 [39]. The actual mean N-input to arable crops 2012–2016 was 124 kg ha$^{-1}$ and nominal reduction targets were assigned as 89 kg ha$^{-1}$ (70%) and 76 kg ha$^{-1}$ (60%).

Sources of nitrogen for crops include applied fertiliser, that mineralised from soil or decomposing plant and animal matter and that deposited from the atmosphere [49]. Crops take up N into the root, stem and grain: that in the roots remains in the soil and contributes N to subsequent crops, whereas the stem and grain are usually harvested and removed from the field (here termed N-crop). Ideally, fertiliser N is applied in such quantity that all sources combine to an equivalent of N-crop. In reality, fertiliser N is commonly applied in excess to cover for possible losses (due to removal in surface wash for example) and in case of yield being higher than expected. The first step, therefore, in assessing the scope for reducing fertiliser N is to estimate the difference between fertiliser input (N-input) and N-crop—a value here termed N-surplus. At the CSC platform, N-input is derived from actual fertiliser application rates. That to the conventional treatment was based on standard agronomic practice. N-input to the integrated treatment was around three quarters of that to the conventional, a value estimated to bring N-surplus close to zero in years of typical crop growth and yield. Beans are usually given no N fertiliser, as here. Dry matter yields were not significantly different between the two treatments except in winter wheat for which yield was mostly below the national average and reduced in the integrated treatment [36]. N-crop is measured directly on above-ground vegetative and reproductive plant material just before harvest.

The government fertiliser survey [39] and yield survey [40,41] were combined to derive estimates of regional-scale fertiliser efficiency and N-surplus. Dry matter in grain was estimated on the basis that yields include 14% moisture content [41]. An indicator—N-input per unit grain dry matter (units kg t$^{-1}$)—was derived to quantify change over 30 years in agronomic efficiency. N-surplus was estimated as N-input from the fertiliser survey minus values of N-crop converted from the yield. First, nitrogen in grain was estimated from dry matter using typical values of %N [50,51], for example 1.4% to illustrate basic trends for all cereals. Nitrogen content of the whole plant was then estimated from a nitrogen harvest index (NHI, N in grain divided by N in the whole plant) typically reported in the literature as between 0.7 and 0.8; the value of 0.77 is used here [51].

## 3. Results

### 3.1. The Challenge: Regional Trends for Grain Crops

Regional data over a 30-year period from 1988 are shown in Figure 2 for nitrogen applied as a mineral fertiliser [39] to the cereal crops (spring barley, winter barley, wheat and oats), and the dry matter yield of these crops [41], together with the nitrogen input per unit yield. The horizontal dashed lines show the respective values around the 1990 base year, averaged over the period of 1988–1992. N-input fluctuated around a mean of 128 kg ha$^{-1}$, reaching a high point of 147 kg ha$^{-1}$ in 2001, but showed no overall trend. In the second half of the period, yields were higher than the base value in all but one year, whereas N-input per unit yield was lower than the base. The main systematic change, therefore, was an increase of around 10% in cereal yield and a corresponding reduction in the amount of N fertiliser needed to support a unit yield.

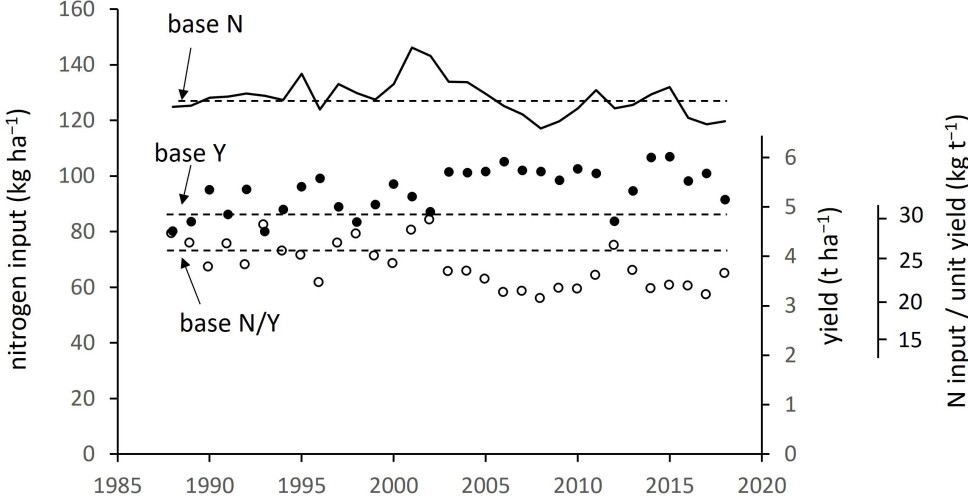

**Figure 2.** Trends for all cereals in nitrogen fertiliser input (N, line), grain yield (Y, closed circles) and N per unit yield (N/Y, open circles), horizontal dashed lines showing the respective base values around the year 1990.

The same primary census data are used in Figure 3 to derive estimates of N in grain and stem, and also N-surplus, assuming typical values of 1.4% N in grain and a 0.77 N harvest index (NHI) as described in the Materials and Methods section. Horizontal dashed lines show the base fertiliser N-input and the nominal 70% and 60% reduction targets. N-input remained well above the 70% target of 89 kg ha$^{-1}$. The combined grain + stem N contents (N-crop) increased in line with the rise in yield in Figure 2.

N-surplus, the unshaded area in Figure 3, was largest at 54 kg ha$^{-1}$ or 38% of N-input in 2002 (arrow i) as a result of high N-input coupled with a reduced yield due to wet weather, and smallest at 7.0 kg ha$^{-1}$ or 6.3% of N-input in 2008 (arrow ii) during a period of the fertiliser price increase that caused a temporary fall in N-input. Over the period of the CSC experiment (2012–2016) N-surplus was estimated at 26 kg ha$^{-1}$, around 20% of N-input. N-crop (grain and stem combined) was higher than the 70% target in most years and higher than the 60% target in all years. This implies that even if N-surplus was reduced to zero under current yield and N content, emissions targets would not routinely be met at a regional scale.

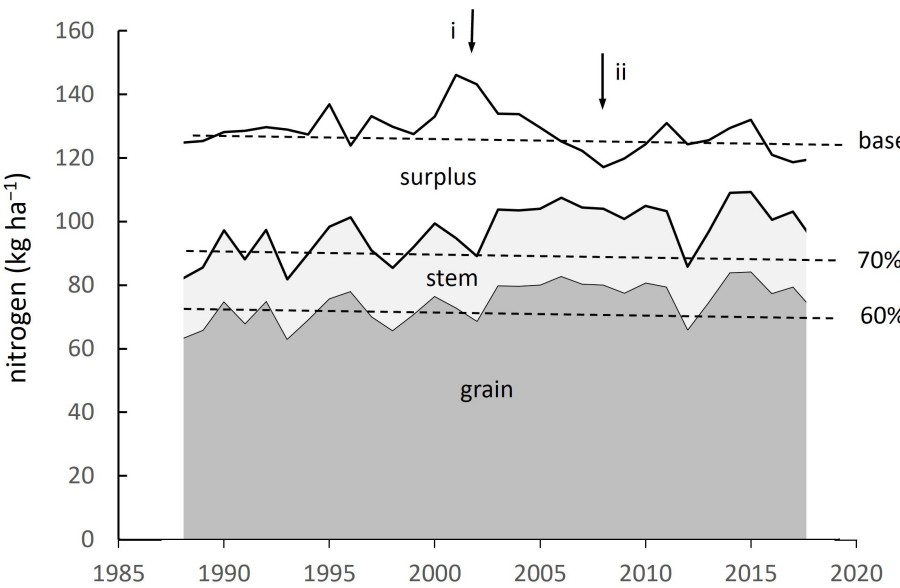

**Figure 3.** Trends for all cereals in nitrogen fertiliser (N-input, upper line) from the annual census [39], and estimated grain and stem nitrogen, based on nominal grain N content and N harvest index (Materials and Methods), and N-surplus (the residual), horizontal dashed lines showing N-input at the 1990 base for emissions, and nominal emissions targets of 70% and 60% of the base; arrows show years of (i) the largest and (ii) the smallest surplus.

The general trends in Figures 2 and 3 should be considered as a context for the experimental studies at the CSC platform. Despite efficiencies in crop management, N-input at the time of the CSC experiment was little different from the 1990 base value, and N-surplus remained positive at all times. The absolute values of N-surplus depend on the assumed values of %N and NHI (neither of which are measured in any yield census); however, the surplus remained positive when calculated over the likely ranges of %N and NHI. The corresponding data for some of the individual cereal crops are based on a small census area, but comparisons indicate N-surplus was smaller in spring than winter cereals, for example 15 kg ha$^{-1}$ for spring barley and 49 kg ha$^{-1}$ for winter barley over 2012–2016. The challenge for the CSC platform is to reduce surplus to near zero in all crops and to devise further solutions for reducing N-input to at least 70% of the 1990 base.

### 3.2. Contribution of the CSC Platform to Reducing Inputs and Surplus

At the platform, N-input in the standard treatment, averaged over the five years (open bars in Figure 4a), was similar to the regional mean in winter wheat and slightly higher than the regional mean for the three other crops. No mineral N fertiliser was given to the bean crop. The mean annual input for three cereals was 154 kg ha$^{-1}$, rising to 168 kg ha$^{-1}$ when oilseed rape was included. These values are higher than the regional average in Figures 1 and 2, but within the range for the mixed system of spring and winter crops (see Materials and Methods). The corresponding average of four crops for the integrated treatment was 126 kg ha$^{-1}$, still well above the regional reduction target of 70% or 89 kg ha$^{-1}$ (Figure 4a). Only integrated SB had an N-input lower than this value. When the bean crop was included in the calculation, the mean N-input was reduced to 134 kg ha$^{-1}$ in the standard treatment and 101 kg ha$^{-1}$ in the integrated treatment.

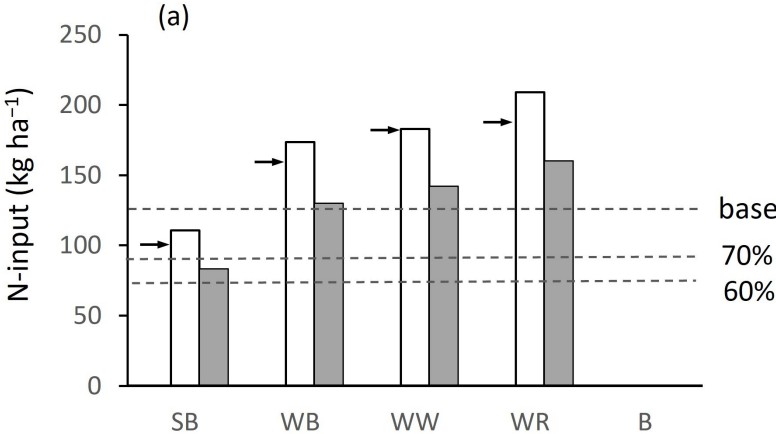

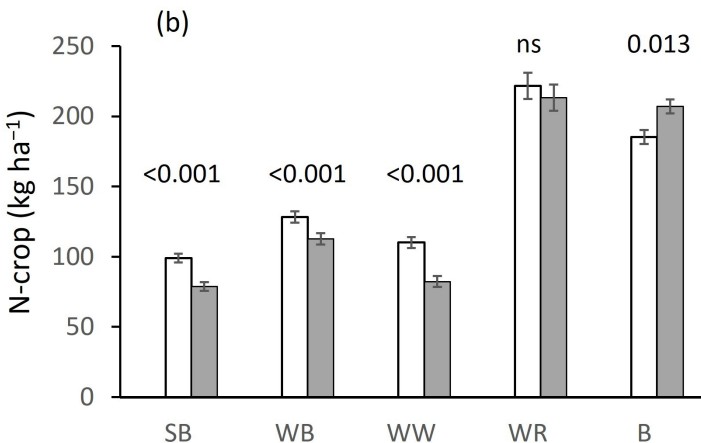

**Figure 4.** Comparison of standard (unshaded) and integrated (shaded) treatments at the platform for SB, spring barley; WB, winter barley; WW, winter wheat; WR, winter oilseed rape; and B, spring bean: (**a**) mineral nitrogen input, arrow indicating mean from regional fertiliser survey, beans not receiving nitrogen, dashed lines indicating regional N-input at 1990 base and reduction targets of 70% and 60%; and (**b**) nitrogen in the plant before harvest, plus and minus SE, *p* values showing significance of difference between treatments.

The uptake of N, quantified as N-crop, was smaller in the integrated treatment, but the relative difference between treatments was reduced (Figure 4b). The gap between treatments had therefore narrowed in the three cereals and had closed in oilseed rape. N-crop in beans was larger than that in any of the cereals, similar to that in oilseed rape, and higher in the integrated than in the standard treatment.

The difference between the respective values in Figure 4a,b determines the N-surplus (Figure 5). The three cereals in the standard treatment had a positive N-surplus, small in spring barley (10% of N-input, but surplus recorded 4 years out of 5) and greater in winter barley (25% of N-input, 5 years out of 5) and winter wheat (39% of N-input, 5 out of 5). The integrated treatment reduced surplus to a little above zero in spring barley (surplus 2 years out of 5), but though smaller, a substantial surplus remained in winter barley (3 years out of 5) and winter wheat (5 out of 5). In contrast, winter oilseed rape returned a mean N gain (the crop took up more than it was given). Averaged across 5 crop-years for the four cereals and oilseed rape, identified as non-legume or NL in Figure 5, N-surplus in the standard treatment was 28 kg ha$^{-1}$, equivalent to 17.2% of N-applied across all four crops, but was reduced in the integrated treatments to 5.6 kg ha$^{-1}$ or 4.4% of N-applied. The grain legume spring bean produced a very large gain: 185 kg ha$^{-1}$ in the standard treatments, 207 kg ha$^{-1}$ in the integrated. When averaged across all 5 crops and 5 years,

the surplus was converted to annual N gain of 17 kg ha $^{-1}$ in the standard and 40 kg ha$^{-1}$ in the integrated treatment. Differences between treatments were largely consistent over the years, but absolute values of N-surplus were largest in the wet 2012 (Materials and Methods) when fertiliser had been applied to the winter cereals, but the yield was much lower than expected. Surplus that year in winter barley, for example, was 107 kg ha$^{-1}$ or 62% of N-input in the standard treatment and 75 kg ha$^{-1}$ or 59% of N-input in the integrated treatment. In contrast, N-surplus in winter barley was low in the high-yielding year 2016: 26 kg ha$^{-1}$ or 14% in the standard and $-12$ kg ha$^{-1}$ (a net gain) in the integrated.

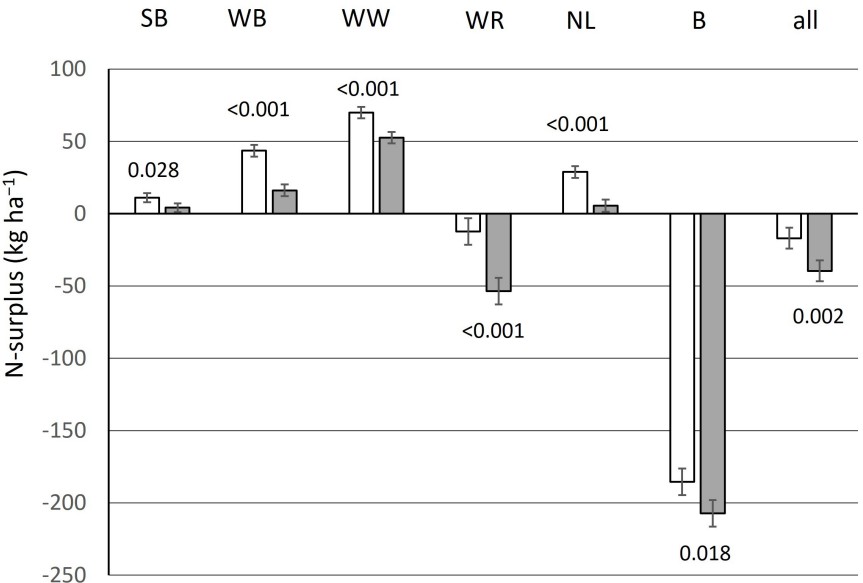

**Figure 5.** Mean N-surplus over five years, negative values indicating crops held more nitrogen than was applied, in (unshaded) standard and (shaded) integrated treatments of spring barley (SB), winter barley (WB), winter wheat (WW), winter oilseed rape (WR), the four non-legumes combined (NL), the grain legume spring bean (B) and all five crops combined (all), plus and minus SE, significance of difference indicated by *p* values.

*3.3. Options for Reaching Nitrogen-Reduction Targets in Main Cropping Systems*

The next aim of the study was to assess whether the reductions of nitrogen fertiliser input at the platform would have the capacity to allow arable cropping to meet reduction targets for the contribution of nitrogen fertiliser to GHG emissions. Potential reductions are now examined for each of the three main cropping systems, defined in the Materials and Methods section as spring, winter and mixed, and the area-weighted mean that combines all three systems. The area-weighted mean N-input (129 kg ha$^{-1}$) was slightly higher than the all-arable 1990 base (127 kg ha$^{-1}$) and all-arable 2012–2016 mean (124 kg ha$^{-1}$) because the all-arable values include some crops that have slightly lower N-input than the cereals.

N-input (2012–2016) based on the regional fertiliser survey is higher than the 70% nominal reduction target in all three systems (Figure 6). To assess the scope to meet targets, N-input in all cereal crops was first reduced by the same percentage as between standard and integrated treatments at the CSC (termed reduced N-input), and then offset further by replacement of 10% and 20% of the main cereals in each system by spring bean or any equivalent grain legume, to which no fertiliser is applied. In the short season, a system consisting mostly of spring barley, the current input was above the 70% target, but reduced input and substitution of 10% legume, each met the 70% target; whereas in combination they met the 60% target. In the mixed system, no target was reached by reduced input or legume substitution alone, but in combination, 10% legume just reached the 70% target, whereas 20% legume reached the 60% target. In contrast, no target was reached in the long season (winter) system. For the area-weighted mean of all three cereal-based systems, the 70% reduction target was not reached by applying reduced input or legume substitution

alone. Targets were only achieved when both were applied together: the 70% target with 10% legume, and the 60% target with 20% legume.

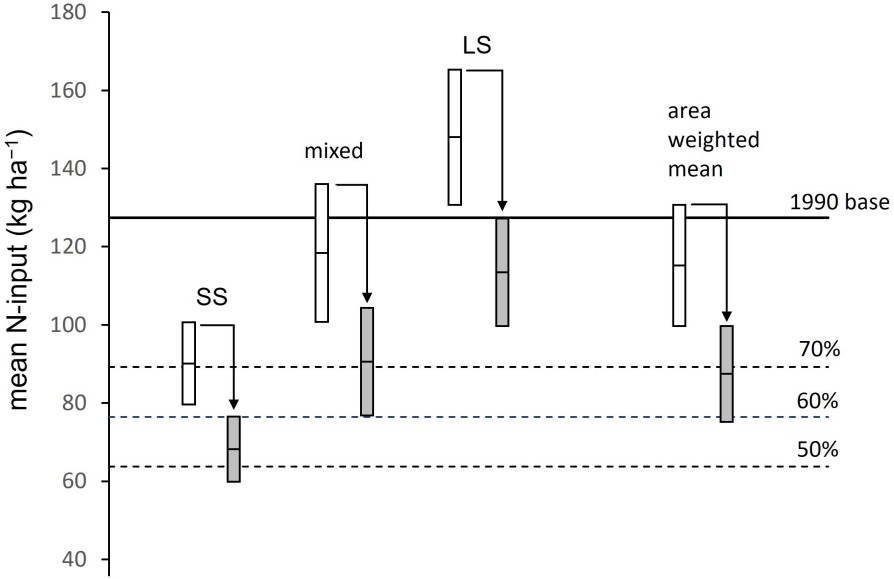

**Figure 6.** Estimated nitrogen fertiliser input to short-season (SS, spring sown), long season (LS, autumn sown) and mixed cereal systems: open bars are for current inputs (2012–2016, based on fertiliser survey) and shaded bars for reduced input comparable to the CSC-integrated treatment; length of bar indicates range from no grain legume (upper), 10% grain legume (mid) and 20% grain legume (lower); the all-arable 1990 base and reduction targets are shown by horizontal lines.

## 4. Discussion

Nitrogen fertiliser inputs to arable cropping in this region have changed little over the past 30 years, remaining well above nominal reduction targets and leaving a substantial surplus not taken up by crops and therefore left to pollute. A systems-level approach to nitrogen management [23] was effective here in demonstrating that major reductions in N-input and N-surplus could be achieved at a commercial field scale. Nitrogen inputs and surplus in the standard treatment were similar to regional means, giving confidence that the platform was representative of arable cropping. The inclusion of two broadleaf crops, one a nitrogen-demanding oilseed and the other a nitrogen-fixing grain legume, reduced net surplus to below zero over five years. Further major reductions in the N-input and N-surplus were achieved through the low nitrification management of the platform's integrated treatment. Notably, in both standard and integrated treatments, the legume produced the largest single annual flux of nitrogen, greater than any quantity given as fertiliser to other crops and originating mostly through biological fixation [37]. Despite improvements, substantial nitrogen loss still occurred in some crops, whereas reduction targets could not be met if the existing areas of cereal, oilseed and legume crops are retained. The main questions, therefore, concern which further interventions are needed to meet zero surplus and emissions targets.

### 4.1. Achieving Zero N-Surplus

Comparisons of platform and regional data confirm the progress towards zero surplus should be possible but reinforced the continued potential to generate surplus. The regional estimates at this period of the CSC first phase, 2012–2016, gave N-surplus for all cereals at around 26 kg ha$^{-1}$ or 20% of N-input. This value is much smaller than the 46% surplus estimated across all agriculture in the UK [31], the main reason being the predominance of spring crops in the region studied, which received less fertiliser to achieve their target yield. They are less prone to limitation by cold, wet weather and, accordingly, their estimated N-surplus is small: 15% estimated from regional data and 10% measured in the standard

treatment at the platform. The management in the integrated treatment at the platform reduced net surplus to near zero when calculated across the three cereals and one oilseed and then raised to a net gain with the grain legume included.

However, the integrated treatment did not eliminate absolute surplus in the cereals, especially in winter varieties. The inherent problem with winter cropping occurred when nitrogen fertiliser had been given in the expectation of a target yield, but that yield was not achieved due to bad weather either restricting growth or operations around harvest or flushing nitrogen out of a field before the crop could take it up. Substantive losses at the platform in 2012 were reproduced throughout the region when the same year gave the lowest cereal yield in several decades, particularly for wheat at 30% below the mean, and a large, estimated N-surplus of 31% of N-input (discernible in Figure 3). The precise fate of the surplus in the winter cereals is uncertain, both at the platform and more widely, but losses are likely through drainage, runoff or conversion to gas before the subsequent crop was able to incorporate nitrogen. Under such circumstances, further genetic improvement in the use-efficiency of nitrogen [51–53] would need to concentrate on capture by roots together with genotypes capable of growing well under adverse conditions. Raising the fraction of nitrogen allocated to grain, which was a major contributor to increasing yield in the 20th century, is unlikely in itself to solve the current problem.

Years of unusually heavy, persistent rain are a feature of the Atlantic zone and are predicted to increase in frequency in some future climatic scenarios [54]. The existing integrated management is likely to raise soil organic matter over time, and hence improve soil structure, microbial activity and nutrient retention, but unless varieties adapted to wet weather are available, further measures may be needed to contain nitrogen when winter cereals were grown. Options trialled elsewhere include inhibiting nitrification by chemical [44] or biological action [24,55] and further amendment by slowly degrading organic material such as biochar [56,57]. Based on the results of the first phase, the CSC management was modified over subsequent years to widen the range of measures to enhance uptake and contain loss of nitrogen. New practices include under-sowing winter cereals with clovers to reduce the need for N-input before winter, direct drilling of seed to minimise soil disturbance, and spatial and temporal adjustment of fertiliser application based on measured soil nutrient balance and crop mass. However, despite these additional measures, crops that are both nitrogen demanding and particularly sensitive to adverse weather—such as the wheat grown in the region—might have to be excluded from soils and localities prone to the effects of excessive rain. The main restriction to achieving zero net surplus routinely is the small area sown with grain legumes, which also affects the attainment of GHG emissions targets, as now discussed.

### 4.2. Achieving Nitrogen Reduction Targets

Given that emissions targets for fertiliser input have not been formally set by governmental bodies, the reductions recommended for the whole of agriculture in Scotland to 70% or 60% of the 1990 base value [34,47,48] were used here as a nominal reference [35]. The platform could directly inform at the scale of mid- to high-input crops over several years, and at this scale, targets could not be met. Even if N-surplus was to be reduced to zero in all crops and years (see previous section), the N required by the long season, autumn-sown plants for growth and grain quality mean that such N-inputs are inevitably higher than the targets. Even the benefit of growing oilseed rape in a winter cereal sequence to reduce surplus runs contrary to the reduction in emissions, since that crop typically receives the highest N-input. Simply reducing inputs with the acceptance of lower yield is also fraught with difficulty, since nitrogen is applied to cereals both to generate the mass of the grain and achieve a specified protein content in the grain. Reducing N-input without reducing the target yield or grain 'sink' could therefore render cereals unfit for their intended markets [58,59].

Nevertheless, the up-scaling via each of the three main cropping systems to a regionally weighted mean (Figure 6) showed that targets could, in principle, be met, but only

through major strategic change to the existing crops and agronomy. The first step would be the adoption of an equivalent to agroecological, integrated or low-nitrification management in all arable fields. Options would need to be tested across the ranges of soil and microclimate to eliminate any unexpected or negative effects. For example, expectations based on individual, controlled studies are not always fully realised at the scale of the cropping system, as concluded for biochar [56]; moreover, some procedures already widely applied, such as the incorporation of crop residues, can have negative effects, in this case by stimulating $N_2O$ emissions [60]. Nevertheless, experience in several regions of the world has shown that progress should be possible by the simultaneous application of a wide range of interventions in field preparation, during growth and after harvest [25].

The second step would be the substitution of some high-input crops by the expansion of nitrogen-fixing grain legumes such as spring bean from the current coverage of 1–2% of the arable surface to a minimum of 10%. As an alternative or addition to grain legumes, short-term grass leys are grown in sequence with spring crops in some areas [38] and could be expanded, but only following a rise in the proportion of forage legumes they contain since current grass leys have a similar N-input to short-season cereals [39], whereas other nitrogen sources used in livestock systems raise total inputs and losses above those for arable [61]. Admittedly, the upscaling in Figure 6 is an approximation that does not account for any subsequent gains or losses of legume N after the crop has been harvested. However, before such secondary effects would accrue, the legumes would still need to be grown in a substantial proportion of fields, and further investigation during such a transition would be needed to maximise benefits to subsequent crops and minimise losses of fixed nitrogen [26]. If grain or forage legumes were increased, a reduction in winter cereal area would be the consequence, with implications for the farm economy and particularly livestock feed for which they are mostly grown [40]. Therefore, an expansion of grain or forage legumes needs to be considered as part of more fundamental change that should also take account of wider issues affecting farming and food security.

### 4.3. Fundamental Change in the Agri-Food System Needed to Meet Emissions Targets

The arable system examined here has qualities typical of enhanced crop production throughout the world. Intensification in the second half of the 20th century allowed the expansion of long-season crops with their higher yield potential but greater sensitivity to yield loss due to inclement weather. In the seven decades since 1960, agriculture here has maintained emphasis on total grain output and output per unit area [32]. The harm due to the accompanying rise in nitrogen fertiliser has not been sufficiently checked or reversed. Similarly, pesticide usage has increased since the 1990s, and high-input and mixed-cropping systems have also suffered a fall in soil quality [45,62] and are more prone to risk of erosion [63].

The future trajectory of arable-grass cropping in the region needs to be redirected to reverse negative trends and at the same time to ensure profitable farming and food security. Questions arise as to whether this could be carried out while maintaining the existing practice of applying mineral nitrogen. This first phase of the CSC showed that the trajectory could be shifted substantially towards lower mineral N usage [24,25]. Ideally, a system currently geared to a production optimum would move towards one satisfying a societal optimum [5]. Such a transition here could be effected in a few years by reducing the area of the long-season winter system and parts of the mixed system (Figure 6) in favour of grain legumes, other short-season spring crops and cereal-legume intercrops, accompanied by under-sowings and catch crops [64]. Actions to reduce N-input should bring wider benefits. For example, replacing some winter oilseed rape with spring varieties, which were prevalent just two decades ago, would immediately reduce nitrogen input and also pesticide input, and reverse some of the decline in in-field food webs [54].

## 5. Conclusions

The findings justify the need for systems research at a field-scale in which crops experience the natural variation in environmental conditions over a period of years. The major contrast between the N-surplus recorded in different years would not have been revealed in a short-term study. Results at the field and half-field scales of the CSC are considered far more representative of commercial agriculture than manipulations in small plots. While the benefits of the CSC have been demonstrated in this first phase, trends in the improvement of soil condition, for example, will take many more years to become effective and are now being tested in subsequent phases. There is increasing worldwide acceptance that major strategic transitions designed to halt losses and pollution are needed, but become possible only if scientific knowledge at a practical field-scale is coupled with socio-economic research to understand the type of incentive needed to bring about effective change [23]. Given the extent of lock-in to existing supply chains, transitions will need reversal of the limited policy support and research funding given to legumes in Europe [29,65] and highly targeted subsidy or credits commensurate with the environmental gains of low nitrification management [3,66].

**Author Contributions:** Conceptualization, G.R.S. and C.H.; methodology, M.W.Y. and C.H.; software, M.W.Y.; validation, G.R.S., M.W.Y. and C.H.; formal analysis, G.R.S., C.H. and M.W.Y.; data curation, M.W.Y.; writing—original draft preparation, G.R.S.; writing—review and editing, C.H.; supervision, C.H.; project administration, C.H.; funding acquisition, G.R.S. and C.H. All authors have read and agreed to the published version of the manuscript.

**Funding:** This work is supported by the Strategic Research Programme of the Scottish Government's Rural and Environmental Science and Analytical Services Division (RESAS): project KUC-F03-2.

**Institutional Review Board Statement:** Not applicable.

**Informed Consent Statement:** Not applicable.

**Data Availability Statement:** Original government data used to generate regional summaries (Figures 1–3) are available for download at the following web sites: crop areas and yield [40,41], fertilizer application [39], rainfall [42]. Data behind Figures 4 and 5 are part of the extensive CSC database and available from C.H. on request. Regional upscaling (Figure 6) is based on data in [38].

**Acknowledgments:** The authors are grateful for assistance from Gillian Banks for laboratory analysis and Andrew Christie for maintenance of the field platform.

**Conflicts of Interest:** The authors declare no conflict of interest. The funders had no role in the design of the study; in the collection, analyses, or interpretation of data; in the writing of the manuscript; or in the decision to publish the results.

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
