# Peer review of "Agroecological Management and Increased Grain Legume Area Needed to Meet Nitrogen Reduction Targets for Greenhouse Gas Emissions"

_nitrogen, doi:10.3390/nitrogen3030035_

Round 1
Reviewer 1 Report
The manuscript tried to argue that agroecological management and increased grain legume area needed to meet nitrogen reduction targets for greenhouse gas emissions. The topic is interesting. However, a few issues need to be resolved.
Major issues include:
1. In the section of Introduction, please present the scientific questions or major objectives, or hypotheses the current study aimed to address. Accordingly, the last paragraph of the introduction section needs to be shortened.
2. The study has large uncertainty. First, lots of data were obtained from literature. How are these data applicable to the region? Second, uncertainties should be considered when conduct the statistics or present the results (e.g., Figure 2). The uncertainties are caused by each variable. Therefore, uncertainty analysis should be conducted.
3. There are no any information about GHG emissions in the method section, but both the Results and Discussion sections made great efforts on GHG emissions.
4. The topic of the manuscript is to show grain legumes are needed to meet nitrogen reduction targets for GHG emissions. However, experiment of elaborate comparison was not available in order to assess whether grain legumes have the advantage in nitrogen input reduction or GHG emission reduction. Similarly, the advantages of grain legumes are largely not supported in the results and discussion sections
Some minor issues include:
Line 30: “effects of nitrogen were known at the start of 20th century intensification in the 1950s”, should be checked.
Line 37: “The global position hardly improved after 1990”, what does it really mean? Please revise.
Lines 187-192: There are several issues in the calculation of N surplus. First, N pool in plants only considered the aboveground parts. Second, N surplus was calculated based on N input and N pool in aboveground plants. How about atmospheric N deposition in this area?
In the Materials and Methods, it is better to put the information of data analysis and statistics as a separate part.
Line 223: “N-surplus is defined as N-input less an estimate of N-crop removed”, please check it.
Line 225, 227 and others: why not directly measure the samples? There are lots of estimates, which increase the uncertainty of the study. For example, the N content of grain should vary spatially depending on soil N status or other factors. A single value of N content should not be possible or suitable.
Line 242: “Regional trends for grain crops” , the cited data should not be used as the results of the current study.
In both the Results and Discussion sections, please check the number of the sub-section titles.
The writing of the manuscript should be improved, and most parts of the manuscript needs to be more concise.
Author Response
The manuscript tried to argue that agroecological management and increased grain legume area needed to meet nitrogen reduction targets for greenhouse gas emissions. The topic is interesting. However, a few issues need to be resolved.
Authors’ response (A-R): We appreciate the time and effort spent by Reviewer 1 in considering the paper, have considered all comments and suggestions and have revised the paper accordingly.
Major issues include:
- In the section of Introduction, please present the scientific questions or major objectives, or hypotheses the current study aimed to address. Accordingly, the last paragraph of the introduction section needs to be shortened.
A-R: The three main aims of the study were enumerated in the final paragraph. However, the main hypotheses to be examined are now stated more explicitly.
- 2. The study has large uncertainty. First, lots of data were obtained from literature. How are these data applicable to the region? Second, uncertainties should be considered when conduct the statistics or present the results (e.g., Figure 2). The uncertainties are caused by each variable. Therefore, uncertainty analysis should be conducted.
A-R. We have incorporated uncertainty where this is feasible. The main contribution of the paper in terms of new data is the field experiment carried out as a split-field design (two treatments) over 5 years with five different crops. (The design, including number of sample sites in the half-fields in each year, is given in Materials and Methods.) There is of course variation among samples and the uncertainty introduced by this variation is handled using standard statistical procedures to estimate the probability that the two treatments differed in attributes such as nitrogen uptake by plants. The results are presented in Figures 4 and 5 which compare the two treatments or combinations of treatments together with standard errors and probability of significance. A general Analysis of Variance was used as a standard statistical technique to summarise this original data in a way that allows us to state whether the two treatments differed. (Most comparisons were in fact significantly different.) Uncertainty is more difficult to incorporate in the analysis of regional context in Figures 2-3. The original source for that context consists of official Government data from only two sources – fertiliser nitrogen input and crop yield, which were collected independently on hundreds of fields but in different surveys. The data represent 25 years of government survey, which we believe offers sufficient certainty that – for example – fertiliser nitrogen inputs have hardly changed. The main uncertainty arises in the estimation of surplus N through assumed values for %N content of cereal grain and the nitrogen harvest index (NHI, nitrogen in grain divided by nitrogen in the above-ground plant). Neither of these attributes were measured in the Government yield survey. Therefore we used values of %N and NHI that were typically cited in the scientific literature (specific sources are referred to). However, we agree that uncertainty in %N and NHI remains a problem. We have therefore modified that section, concentrating on the basic data (nitrogen input and yield) for all cereals combined, and have removed the less certain analysis of spring and winter crops.
- There are no any information about GHG emissions in the method section, but both the Results and Discussion sections made great efforts on GHG emissions.
A-R. We accept the possible confusion here and have amended text in the Introduction and Materials and Methods to show the approach taken. This paper does not estimate GHG emissions directly, but refers to standard Government estimates based on regular survey and analysis across all industrial and domestic sectors. Governments compute GHG emissions based on agreed upscaling factors, such as those that a typical car, livestock unit or fertiliser application (for example) might generate in a given period. The main emissions from agriculture are from livestock and arable, and the main contributary factor in the latter is nitrogen fertiliser, both in its manufacture and usage in the field. The background and reason for our study arise from the UK Government’s analysis that agriculture in Scotland, and in particular arable cropping, has not achieved adequate reductions since the base year of 1990 (paper cited: CCC 2018). Government recommends a reduction in agriculture to at least 70% of the 1990 emissions estimate but does not specify where that reduction should occur (e.g. in certain agricultural sectors or evenly across the board.) In the revised Materials and Methods, we provide background by citing examples of CO2-equivalents for agriculture, but then concentrate on options to reduce nitrogen input to 70% or lower of the 1990 base. We feel that agriculturalists will find more practicable to appreciate reductions in terms of units of nitrogen input rather than CO2-equivalents. We have also clarified text in the Discussion and have replaced terms such as ‘GHG emissions targets’ with ‘nitrogen reduction targets’.
- The topic of the manuscript is to show grain legumes are needed to meet nitrogen reduction targets for GHG emissions. However, experiment of elaborate comparison was not available in order to assess whether grain legumes have the advantage in nitrogen input reduction or GHG emission reduction. Similarly, the advantages of grain legumes are largely not supported in the results and discussion sections
A-R. We accept that the Discussion might have generalised to ‘grain legumes’ in preference to naming the grain legume used in the main experiment, which was field bean, Vicia faba. The results of the main experiment clearly show that field bean – a grain legume - fixes very large amounts of nitrogen and in doing so converts an overall nitrogen loss to a small nitrogen gain, i.e. more in the plants than applied as fertiliser (Fig. 5). Field bean typically receives no nitrogen fertiliser and so substituting it into a cereal-based system reduces overall N inputs. The only other option is to introduce a year of fallow (no crop, e.g. the practice of set aside in the 1990s), but that would reduce overall output whereas field bean or another grain legume provides a commercial output. We have therefore clarified text in the Results and Discussion sections.
Some minor issues include:
Line 30: “effects of nitrogen were known at the start of 20th century intensification in the 1950s”, should be checked.
A-R. Checked and clarified
Line 37: “The global position hardly improved after 1990”, what does it really mean? Please revise.
A-R. It has been clarified
Lines 187-192: There are several issues in the calculation of N surplus. First, N pool in plants only considered the aboveground parts. Second, N surplus was calculated based on N input and N pool in aboveground plants. How about atmospheric N deposition in this area?
A-R. We agree that a full nitrogen balance study would need to include soil-N sources, N accumulation below ground and N-deposition, but to fulfil its aims, this paper concentrates on the reduction of nitrogen fertiliser input. Whatever the value of N deposition, agriculture will need to reduce fertiliser N if it is to meet nitrogen reduction targets. The text has been amended.
In the Materials and Methods, it is better to put the information of data analysis and statistics as a separate part.
A-R. The statistical analysis is carried out only on the CSC experiment, not on the preceding contextual data. It might be preferable therefore to leave the short paragraph on statistics in the experimental section.
Line 223: “N-surplus is defined as N-input less an estimate of N-crop removed”, please check it.
A-R. For explanation please refer to the Authors’ response to the reviewer’s comment above re. Lines 187-192
Line 225, 227 and others: why not directly measure the samples? There are lots of estimates, which increase the uncertainty of the study. For example, the N content of grain should vary spatially depending on soil N status or other factors. A single value of N content should not be possible or suitable.
A-R. As discussed in relation to the second major point above, the government data we used to set the context came from two sources only – fertiliser and yield survey. These surveys do not include nitrogen content of grain or any within-field spatial measurements. In fact there are no published large scale surveys that measure %N for example.The revision defines more explicitly the two sources of data and their limitations, and (as discussed above) revises Fig. 2 and Fig. 3 to concentrate on main trends without separating out crop types.
Line 242: “Regional trends for grain crops” , the cited data should not be used as the results of the current study.
A-R. To provide the context, the paper presents original government survey on nitrogen input and yield (with references), but then goes further to combine the data sets to generate results that are not published in any of the government surveys (Fig. 2, 3) and which therefore become results of the current study.
In both the Results and Discussion sections, please check the number of the sub-section titles.
A-R: all checked
The writing of the manuscript should be improved, and most parts of the manuscript needs to be more concise. A-R. We try to follow the underlying approach in the various updates of the English language style guide “Plain Words“. However, we have gone through the text, clarifying and reducing it where necessary.
Reviewer 2 Report
The manuscript deals with the responses of grain yields and greenhouse gas emissions to agroecological managements at regional scale. The manuscript is generally well written and easy to follow. I have several minor suggestions that the authors may consider.
1. section 2.3, the calculation of GHG emissions is lacking. It would be good to readers that the authors give a brief introduction rather than merely references.
2. section 2.2, the low-nitrification managements are confusing. The authors may clarify how these managements worked.
Author Response
The manuscript deals with the responses of grain yields and greenhouse gas emissions to agroecological managements at regional scale. The manuscript is generally well written and easy to follow. I have several minor suggestions that the authors may consider.
Authors’ response. We appreciate the time and effort taken by Reviewer 2 to read and comment on the manuscript. We have made changes as follows
- section 2.3, the calculation of GHG emissions is lacking. It would be good to readers that the authors give a brief introduction rather than merely references.
Authors’ response. We accept the possible confusion here and have amended text in the Introduction and Materials and Methods to show the approach taken. This paper does not estimate GHG emissions directly, but refers to standard Government estimates based on regular survey and analysis across all industrial and domestic sectors. Governments compute GHG emissions based on agreed upscaling factors, such as those that a typical car, livestock unit or fertiliser application (for example) might generate in a given period. The main emissions from agriculture are from livestock and arable, and the main contributary factor in the latter is nitrogen fertiliser, both in its manufacture and usage in the field. The background and reason for our study arises from the UK Government’s analysis that agriculture in Scotland, and in particular arable cropping, has not achieved adequate reductions since the base year of 1990 (paper cited: CCC 2018). Government recommends a reduction in agriculture to at least 70% of the 1990 emissions estimate but does not specify where that reduction should occur (e.g. in certain agricultural sectors or evenly across the board.) In the revised Materials and Methods, we provide background by citing examples of CO2-equivalents for agriculture, but then concentrate on options to reduce nitrogen input to 70% or lower of the 1990 base. We feel that agriculturalists will find more practicable to appreciate reductions in terms of units of nitrogen input rather than CO2-equivalents. We have also clarified text in the Discussion and have replaced terms such as ‘GHG emissions targets’ with ‘nitrogen reduction targets’.
- section 2.2, the low-nitrification managements are confusing. The authors may clarify how these managements worked.
Authors response. Previous authors (e.g. Robertson et al. 2013; Norton and Ouyang 2019) have promoted an approach to low nitrification management that includes a range of practices (termed a portfolio approach) – the reason being that the underlying processes are highly complex, vary with climate, soil and crops, to an extent that no single practice is likely to be successful. We have therefore clarified the text in 2.2 to show how the various agronomic changes we imposed might each contribute to reduce loss of nitrogen.
Round 2
Reviewer 1 Report
The issues I presented were well addressed.